# Differences in Wear and Material Integrity of NAO and Low-Steel Brake Pads under Severe Conditions

**DOI:** 10.3390/ma14195531

**Published:** 2021-09-24

**Authors:** Edouard A. T. Davin, Anne-Lise Cristol, Arnaud Beaurain, Philippe Dufrénoy, Neomy Zaquen

**Affiliations:** 1Univ. Lille, CNRS, Centrale Lille, UMR 9013-LaMcube-Laboratoire de Mécanique, Multiphysique, Multi-échelle, F-59000 Lille, France; anne-lise.cristol@centralelille.fr (A.-L.C.); arnaud.beaurain@univ-lille.fr (A.B.); philippe.dufrenoy@polytech-lille.fr (P.D.); 2LAPINUS, ROCKWOOL BV, Delfstoffenweg 2, 6045 JH Roermond, The Netherlands; neomy.zaquen@lapinus.com

**Keywords:** friction materials, wear, integrity, high temperature, fibers

## Abstract

In this study, through severe reduced-scale braking tests, we investigate the wear and integrity of organic matrix brake pads against gray cast iron (GCI) discs. Two prototype pad materials are designed with the aim of representing a typical non-metal NAO and a low-steel (LS) formulation. The worn surfaces are observed with SEM. The toughness of the pad materials is tested at the raw state and after a heat treatment. During braking, the LS-GCI disc configuration produces heavy wear. The friction parts both keep their macroscopic integrity and wear appears to be homogeneous. The LS pad is mostly covered by a layer of solid oxidized steel. The NAO-GCI disc configuration wears dramatically and cannot reach the end of the test program. The NAO pad suffers many deep cracks. Compacted third body plateaus are scarce and the corresponding disc surface appears to be very heterogeneous. The pad materials both show similar strength at the raw state and similar weakening after heat treatment. However, the NAO material is much more brittle than the LS material in both states, which seems to favor the growth of cracks. The observations of crack faces suggest that long steel fibers in the LS material palliate the brittleness of the matrix, even after heat damage.

## 1. Introduction

Friction materials used for commercial automotive brake pads mostly belong to the family of non-asbestos organic (NAO) materials. These materials usually consist of a resin matrix, various types of fibers, filler components, as well as friction and wear modifiers. These formulations can include metallic constituents such as iron or copper. NAO formulations containing a small amount of steel fibers are called low-steel (LS) materials.

In addition to choices based on technical characteristics, the choice to use simple NAO or LS materials is geographically oriented depending on the continent, due to customer preferences and regulations. However, due to increased market demand, there is a growing need for a global concept. In the meantime, there has been a tendency to reduce the metal content in brake pads.

The braking contact is composed of two first bodies (disc and pad) and an intermediate layer, called the third body [1]. The third body performs three main functions: bearing contact loads, accommodating speeds, and ensuring separation of the first bodies. The third body particles travel on the surface of the first bodies and form the tribological circuit [2,3]. Various flows of particles are likely to be activated in this circuit, in particular:Material that detaches from the first body constitutes the main source flow;Particles that circulate inside the contact while accommodating velocities constitute the internal flow;Particles that leave the contact permanently constitute the wear flow.

Primary and secondary contact plateaus form at the interface [4]. The primary plateaus consist of the wear resistant constituents of the pad (such as steel fibers or ceramic particles) and form nucleation sites for the secondary plateaus, which are formed by compaction of the detached particles in front of the primary plateaus.

The use of steel fibers is known to improve the performance of several aspect of brake pads such as helping to remove heat from the contact interface [5,6] and creating primary contact plateaus [7,8]. Although several studies have reported that introducing steel fibers or increasing their content led to high wear of the contact [9,10,11], other studies have shown mixed results regarding this matter [6,12,13]. Non-metal NAO formulations are often limited to low or moderate temperature use, due to the degradation of the organic resin matrix and aramid fibers which happens between 200 and 600 °C [14,15,16,17].

The wear and integrity of brake components under severe conditions have been investigated in various studies. Some researchers have focused on very specific types of environmental conditions [18], while many researchers have studied cases of high energy and emergency braking. Most of these studies have focused on the loss of integrity by the apparition of cracks in gray cast iron discs (such as [19,20]) and steel discs [21,22] and have highlighted the role of alternating tensile-compressive stress cycles favored by high temperature gradients and rapid temperature changes in the disc.

Fewer studies have investigated the excessive wear and loss of integrity of brake pads or shoes for drum brakes. Some studies have focused on metal-matrix brake pads (such as [23]), designed for high-speed trains or heavy-duty vehicles. Regarding organic matrix brake materials, Barros et al. [24] highlighted the role of contact pressure in the transition to severe wear. Laguna-Camacho et al. [25] studied drum shoes used in real service for eight months in a warm environment and found major cracks that they attributed to fatigue from repeated impacts of the shoe against the drum. Elzayady and Elsoeudy [26] studied three formulations of NAO pads in real service and in severe laboratory braking tests, and found cracks attributed to thermal fatigue. They noted that the addition of graphite and barite lowered the generated heat and produced fewer cracks. Park et al. [27] studied the wear and airborne emission with several low-steel and non-steel pad formulations and found that the low-steel formulation generated more wear and more airborne particles. They also noted that more airborne particles were emitted at higher velocities. Their fine surface observations enabled them to draw schematics of the pads’ surface topographies. Cong et al. [15] focused on the role of phenolic resin in their tests at high speed and moderate load. A pad material using their new resin modified with latex generated about 25% less wear than a pad with classic resin. It also retained its integrity while the classic material suffered millimetric cracks.

As a first step towards a global concept for improved formulations, in the present study, therefore, we propose to study the mechanisms responsible for differences in wear and integrity between non-metal NAO formulations (simply referred to as “NAO” in this article) and LS formulations. As temperature seems to be a major discriminant that limits the use of NAO materials, we choose to study high-temperature behavior. A specific focus is placed on the role of mineral and steel fibers, which represents a major difference between the two types of formulations.

We conduct a reduced scale experimental laboratory study with two prototype formulations, designed by the authors, aimed at representing typical NAO and LS materials. The pads are slid against gray cast iron discs. The braking test data are studied and friction surfaces are observed with high resolution photography and scanning electron microscopy (SEM) and analyzed with energy-dispersive spectroscopy (EDS), in order to investigate the involved phenomena. Wear is monitored, as well as markers of the integrity of the friction parts. An interpretation of the tests results is conducted with the help of the mechanical characterizations of the pad materials. Taken together, these results are discussed to propose links between the observed phenomena and the properties of the material formulations. The use of a reduced scale tribometer allows for very close monitoring and detailed understanding of the phenomena involved at the interface. Unlike general-purpose standard braking tests, the design of a tailored test program, with a focus on scientific understanding, allows us to keep only a limited set of desired parameters and a well-controlled solicitation range. Coupling such tests with multiscale observations and analyses of worn surfaces, with observations of crack faces, as well as mechanical characterizations of the materials constitutes a novel approach. This approach covers a continuum of brake system performance, tribological interface behavior, mechanical behavior of the parts, and the design of the material formulations.

## 2. Materials and Methods

### 2.1. Friction Materials

The materials of the friction parts aim at representing typical braking with NAO and LS pads. The discs are made of grey cast iron (GCI) EN-GJL-250, with yield strength of 250 MPa (as per the standard) and measured hardness of 180 HV. The material configurations are referred to as NAO-GCI (NAO pad against gray cast iron disc) and LS-GCI (LS pad against gray cast iron disc).

The pad formulations are composed of 10 constituents (LS) and 12 constituents (NAO), which are detailed in Figure 1. The raw constituents are classified under four generic categories (fibers, binder, fillers, and friction and wear modifiers) to ease understanding, but one has to keep in mind that each raw material leads to several functions, and that some functions arise from the interactions among several types of constituents. Volume fractions are chosen as they can be compared with surface observations more easily than mass fractions.

In volume fractions, the two formulations show many similarities. One of the key questions in this study is the role of the fibers. The LS formulation used steel fibers, while the NAO formulation used Lapinus short mineral fibers, namely RB215ELS. As shown in Table 1, the two types of fibers are very different in size. This difference is mainly due to the manufacturing processes for steel and mineral materials. Other than the fibers, some constituents (para-aramid pulp, phenolic resin, barite, friction particles, and Promaxon-D) were the same between the two formulations, while some constituents were changed (in type or amount). These choices, driven by the results of preliminary braking tests, were made in order to obtain two functional pad materials.

Before the braking tests, the pads were scorched for 150 s against a 450–500 °C metal plate, as previous tests have shown this procedure provided a steadier coefficient of friction (CoF).

The microstructure of both formulations is presented in Figure 2. The material of the NAO formulation is not homogeneous at this millimetric scale, mainly due to the presence of a small amount of large graphite particles. The presented area was selected to display such graphite particles. The material of the LS formulation is more homogeneous.

The diffusivity values of both pad materials were measured by means of laser flash analysis (LFA) along the direction normal to the pads’ surfaces. The diffusivity values are given in Table 2.

### 2.2. Braking Tests

Before starting the study, several types of full-scale braking tests were conducted with the chosen materials. The test that was better able to discriminate the friction materials (regarding wear, integrity, and friction) was chosen for study at a reduced scale.

The scale change was done by performing a thermal similitude that kept the same thermal flux in the friction parts at both scales [28,29]. The strategy of reducing the scale made the experiments easier to monitor, with smaller sliding surfaces that are easier to access for observation. Moreover, it allowed for a simplified braking system which provided better control of the system parameters. For these reasons, the reduced scale tests enabled a better understanding of the involved interface phenomena.

The full-scale braking tests are not reported in this article. The selected program for the reduced scale tests (defined in Table 3) was derived from a standard high-energy braking test called high-speed fade (HSF). The program was simplified to keep only a run-in sequence, a single high-speed braking, and a high-temperature braking sequence. During the high-temperature sequence, disc temperature increased from 250 to 500 °C. This temperature increase was controlled with a set of given initial disc temperatures in order to reproduce the temperature rise of the full-scale tests.

The reduced scale laboratory tribometer (shown in Figure 3) is an inertial bench, which is an evolution of the tribometer described in [29,30]. The disc is vertical, rotated by an electrical motor and features a one-sided contact with a pad. The normal contact force was applied on the pad by a motorized linear actuator through a loading spring. The use of a deformable structure ensured a linear motion without any free play. The brake pad was linked to the structure through a three-directional piezoelectric force sensor, which enabled evaluation of the contact forces in the normal, circumferential, and radial directions. The apparent contact pressure was calculated by dividing the normal force by the apparent contact area of the pad. The friction coefficient was obtained by dividing the friction force by the normal force, and applying a rolling average filter with window width of 0.5 s. During a braking, the normal load was controlled at a constant force.

The tribometer can simulate the braking deceleration of a vehicle, taking into consideration inertia and braking torque. The vehicle’s inertia is simulated with a flywheel mounted on the shaft and a torque component from the motor. The initial speed is set at the beginning of the braking. Then, during the braking, the speed is adjusted with an instantaneous deceleration, calculated as the ratio of the instantaneous braking torque by the simulated inertia.

A non-contact position sensor monitors the travel of the brake pad, and thus allows for continuous evaluation of contact wear. The very high stiffness of the force sensor ensures that its deformation is negligible as compared with the values of worn thickness. The values of pad travel from the sensor were verified with thickness measurements of the friction parts before and after the test. The travel signal was processed by a median filter with window width of one second, followed by a rolling average filter with window width of one second. This filtering process was repeated three times. Then, the time derivative of this filtered signal was approximated by finite differences in order to evaluate the wear rate.

Both friction parts featured specific geometries, as shown in Figure 4. Thermocouples were used to measure the temperature at specific points in the parts. Three thermocouples, i.e., TP1, TP2, and TP3, were inserted in the pad, along its centerline, 3.6 ± 0.1 mm from the surface. One thermocouple, TP4, was located near TP2, but 7.5 ± 0.1 mm from the surface. One thermocouple, TD, was inserted at the middle of the disc track, 1.8 ± 0.15 mm from the surface.

### 2.3. Investigation of Cracked Parts

When friction parts undergo cracks as a result of braking conditions, we aimed to study the phenomena leading to this loss of integrity. Figure 5a shows the procedure used to open the existing crack. A slit was machined on the back face of the pad, opposite the crack, and then a small bending moment was applied in order to propagate the crack and break the pad. Then, the crack face could be observed, as shown in Figure 5b. Figure 5c shows how milling and polishing enabled observation of the material in the vicinity of the crack.

### 2.4. Mechanical Tests

Pad materials were tested in three-point flexure, with beams having a rectangular cross-section. Figure 6 shows the pattern for sampling either two flexure test beams or one reduced-scale friction pad from the original lump of molded material. The measured section of the beams was approximately 15 × 15 mm² (varying with samples), the flexure span was 50 mm, and the crosshead speed was 2 mm/min. The upper and lower surfaces of the beams were machine ground and softly prepared with fine grained sandpaper in order to avoid large geometric defects. When analyzing the data, the maximum stress and strain at the outer surface (at mid-span), as well as the corrected zero-deflection point, were calculated following the ASTM standard D 7264/D 7264M [31]. The test was stopped by the operator after the beam was broken.

One should keep in mind that the sample geometry changed drastically after the stress peak (apparition of crack), and therefore the values of stress and strain, calculated with initial geometry, could not be considered quantitatively after this point.

In order to investigate the evolution of material toughness when exposed to high temperatures encountered during braking, some beams were exposed to heat before the flexure test (as a thermal pretreatment). It was not possible to extract samples from rubbed pads due to the heterogeneity induced by the thermal gradient and the presence of many cracks. For the treatment, the beams were placed in a 600 °C oven, for 20 min with air atmosphere. Then, they were cooled in ambient air, and submitted to the flexure test when they reached room temperature. For each material, one beam was tested without treatment, and one was tested after thermal exposure.

## 3. Results

### 3.1. Braking Tests

Figure 7 shows the experimental data extracted from a test with the LS-GCI configuration; only a part of the test is displayed, including the last two brakings of the run-in sequence, the single high-speed braking, and the first ten high-temperature brakings. Data were only plotted during pad-disc contacts. Figure 7a shows the sliding speed and the apparent contact pressure against time. Figure 7b shows an estimation of the CoF and the wear rate of the contact (in thickness per unit of time), and Figure 7c shows temperatures measured in the disc and in the pad at several points. Variations of the pressure (mainly in the range 1.9–2.3 MPa) give clues about the occurrences of contact vibrations.

At the end of the run-in sequence, the CoF is in the range of 0.35–0.40, and the wear rate is too low to be measured by our setup. All pad temperatures are roughly above 50 °C, and disc temperatures are roughly above 100 °C.

During the high-speed braking, the CoF drops to 0.25, and then increases slightly at the end of the braking, while the wear rate is 9 µm/s at the beginning, and decreases roughly proportionally to speed. The temperatures in the pad near the surface increase from 50 °C to a maximum of 205 °C at the outlet of the contact. As expected, the temperature further from the surface increases slower and with a delay, and reaches 100 °C.

During the high-temperature sequence, the CoF drops to the range of 0.22–0.27 at the first braking, and then increases steadily until it nearly comes back to its run-in value at Brakings 9 and 10. The wear rate remains roughly proportional to the sliding speed during each braking. Its initial value increases for five breakings, up to about 30 µm/s. Then, it decreases slightly and stabilizes just above 20 µm/s. Disc temperature increases from 250 to 500 °C as commanded by the temperature program. The pad temperatures near the surface increase from 175–200 °C (beginning of the first braking) to 430–530 °C (end of the tenth braking). In the meantime, the temperature further from the pad’s surface increases from 120 to 310 °C. All measured temperatures tend towards a steady state after 8–10 brakings. One should note that the worn thickness of the pad is not negligible. For instance, after ten high-speed brakings, all pad thermocouples are located about 1.1–1.2 mm closer to the pad surface than they were before wear.

Figure 8 shows the experimental data extracted from a test with the NAO-GCI configuration. One should note that the time axis values are not the same in Figure 7 and Figure 8, despite running the same test program. This is due to the fact that the brakings are not controlled by a fixed duration, but with a given amount of initial kinetic energy. Therefore, the duration of each braking depends on the actual friction coefficient. The time between brakings also depends on the system’s cooling time, which varies with the temperature attained.

At the end of the run-in sequence, the CoF is in the range of 0.25–0.30, and the wear rate is too low to be measured by our setup. All pad temperatures are roughly above 50 °C, and disc temperatures are roughly above 100 °C.

During the high-speed braking, the CoF starts around 0.25–0.30, and then suddenly drops to 0.1 and increases slightly at the end of the braking. The wear rate increases and peaks at 80 µm/s, and then decreases sharply. The temperatures in the pad near the surface increase from 50 °C to a maximum of 165 °C at the outlet of the contact. As expected, the temperature further from the surface increases slowly and with a delay, and reaches 70 °C.

The high-temperature sequence could not be carried out entirely, due to large contact wear exceeding the translation motion of the normal force actuator. This effect appears clearly with a sharp drop in contact pressure during the fifth high-speed braking. Therefore, only these five brakings are studied further. The CoF quickly increases and exceeds its value from the run-in sequence. The wear rate always increases at the beginning of the braking, and then peaks and decreases sharply. Its initial value varies between 40 and 60 µm/s. Disc temperature increases from 250 to 500 °C, as expected. Pad temperatures near the surface increase from 150–170 °C (beginning of the first braking) to 405–505 °C (end of the fifth braking). In the meantime, the temperature further from the pad’s surface increases from 90 to 220 °C.

The analysis of the test data with both configurations can be summarized as follows:
At the end of the run-in sequence, the CoF with the LS-GCI configuration is 30–40% higher than the CoF with the NAO-GCI configuration.During the high-speed braking sequence, for both configurations, the CoF drops suddenly, and then increases.With the LS-GCI configuration:
○The wear rate seems to depend on the sliding speed;○The wear rate increases during the high-speed and high-temperature braking sequences;○The wear rate reaches a steady state during the high-temperature braking sequence.
With the NAO-GCI configuration:
○The wear rate is far higher than with the LS-GCI configuration (up to about 6–8 times as much, confirmed by post-mortem thickness measurements);
○The wear rate shows large variations among the brakings;
○The wear is so high that the test session cannot be carried to completion;
○The temperature rise in the disc varies significantly from one braking to another.




### 3.2. Surfaces Analysis

Figure 9 shows photographs of the worn parts. Figure 9a,b displays the pad and disc surfaces, respectively, of the LS-GCI configuration; both surfaces look even in the sliding direction, but differences of shades and color appear in the radial direction. Overall, the integrity of the surfaces is very good, as no large damage can be seen. Figure 9c shows a side view of the NAO pad at its inner radius. Figure 9d,e displays the pad and disc surfaces, respectively, of the NAO-GCI configuration. The surface of the NAO pad features a dozen macroscopic cracks perpendicular to the sliding direction, and the sideview gives a hint about their depth of several millimeters. On the worn surface, coarse graphite particles remain visible, which highlight the macroscopic heterogeneity of the pad formulation. The image of the corresponding disc shows large variations of aspect, with an even area on the left side, and stripes in the direction of sliding, forming a dark patch on the right side. This observation matches the hot spots seen by thermography during the braking (data not presented here).

The dramatic wear of the NAO-GCI configuration, as well as its erratic behavior, confirm that high temperature is indeed a challenge when using the NAO pad formulation, when the LS-GCI reaches a steady state and ends the test successfully. In order to investigate this behavior further, we observe the worn surfaces under SEM at a local millimetric scale, which is able to investigate tribological mechanisms.

Figure 10 shows an SEM micrograph of a worn LS pad. The surface is mostly covered by a solid layer. Graphite particles can be seen because they are not adhesive.

Figure 11a shows a closer view of the solid layer at the surface of a worn LS pad. EDS analyses were conducted on six rectangle areas, labeled from 1 to 6. The EDS spectra for the six areas are very similar both qualitatively and quantitatively, so much so that most spectra are nearly indistinguishable from one another. For this reason, only one spectrum is presented in Figure 11b. The spectrum shows large peaks for Fe and for oxygen. A low peak of carbon is visible, while Ca, Ba, and S appear in very low amounts. Oxidized iron is common in braking interfaces; however, it is usually found in lose particles or compacted powder. Here, it is in the form of a solid layer. It is consistent with the previous observation, i.e., with the explanation that the very high temperatures melted (or strongly sintered) the metal, leaving this homogeneous solid layer.

Figure 12 shows a SEM micrograph of a worn NAO pad. The surface displays a gray layer, spotted with light and dark particles that are rather coarse as compared with usual third body particles. This observation suggests that this layer was created recently, and did not have much time to be finely ground in the contact. Some plateaus of compacted third body look smoother and more compacted than the surrounding material. The third body layer shows several microscopic cracks. The area is crossed by a large crack that looks very deep.

Figure 13a shows a closer view of the surface of a worn NAO pad. The EDS analyses were conducted on three rectangle areas labeled as b, c, and d, in Figure 13a. The corresponding EDS spectra are shown, respectively, in Figure 13b–d. The spectra are cropped to 7.5 keV as no peaks appeared beyond that value. The spectrum in Figure 13b displays a large peak of Si which indicates the presence of either a mineral fiber or debris from Promaxon-D. The peak around 3.67 keV may be attributed to either Ca or Sn, or the sum of both. Ca would originate from a mineral fiber, while Sn would come from metal sulfides. Other elements are in low amounts. Some of these elements originate from the pad formulation, such as K and Ti (from potassium titanate), Zr (from zirconium dioxide), Al (from mineral fibers), C (from graphite), and S (from metal sulfides). Since Fe is found in small amounts, it may originate either from the pad’s metal sulfides or from the disc’s cast iron. The oxygen may originate either from native pad constituents or from the oxidation of the materials by the surrounding air. The spectrum in Figure 13c displays large peaks of Ba, S, and O, which indicate the major presence of baryte. The smaller peaks of Si, Ca, K, Zr, Al, Fe, and C can be explained by the same reasoning as in the spectrum in Figure 13b. The spectrum in Figure 13d displays a large peak of Si, indicating material from a mineral fiber or from Promaxon-D. The peak at 3.67 keV indicates either Ca or Sn, or both, as previously mentioned. Most other peaks are similar to the ones found in the two previous spectra, with varying heights. One can note the presence of Mg, found in the mineral fibers. The increased presence of Fe may indicate material coming from the disc.

These analyses reveal that the raw surface of the material remains mostly visible, with little third body, probably resulting from an aborted tribological circuit, i.e., without sufficient accumulation of debris to constitute a third body that is thick enough to protect the material and prevent excessive wear. More considerations on this interpretation are detailed in the discussion.

In order to better understand the appearance of cracks in the NAO pad, we observed their face. Figure 14a shows one side of the face of a crack that had been opened following the procedure described in Section 2.3. Figure 14b shows the same place after the surface had been machined and polished to display the material in the vicinity of the crack; the crack face looks rather rough. The hole of a thermocouple can be seen near the crack. The presence of this hole may have favored the creation of the crack, but it was probably not decisive, as the hole is not exactly on the crack’s path, and most cracks appeared in areas devoid of any hole. The crack face shows a gradient of color, with a dark grey tone close to the friction surface that transitions to a light brown at a few millimeters of depth. This gradient may be due to a change in the material, or to a coloration of the surface, for instance, due to the deposition of worn particles. The milled and polished surface in Figure 14b shows a similar color gradient. This confirms that the gradient appears on the crack face, and also deeper in the material. Therefore, we suggest that this color gradient traduces a modification of the material close to the friction surface. Finer analyses of this crack face were conducted with SEM and EDS, but did not provide any better understanding of the potential change in the material.

The analysis of surfaces can be summed up by the following points:
With the LS-GCI configuration:
○The friction surfaces look even and feature good integrity;○The pad is mostly covered with a layer of solid oxidized iron, which looks like it was melted during braking.
The friction surfaces look even and feature good integrity;
○The aspect of the disc varies significantly along the circumference;○The pad shows several multi-millimeter-deep cracks;○At a millimetric scale, the pad displays coarse particles of native constituents, and only a small amount of fine-grained compacted plateaus;○The pad material displays a gradient of color close to the friction surface, interpreted as a modification of the material during braking.


### 3.3. Mechanical Tests

The mechanical properties of the pad materials are tested following the procedure described in Section 2.4. Data of maximum stress and strain in three-point flexure (at the outer surface, at mid-span) are plotted for each material and for both states (raw material and material exposed to 600 °C in air atmosphere for 20 min, and then cooled), as shown in Figure 15.

All plots show a toe region at the beginning attributed to the take-up of slack, the alignment, and the seating of the sample, as suggested by the ASTM standard D 7264/D 7264M [31]. Then, a somewhat linear behavior appears, followed by a peak that represents ultimate strength (US), and a decrease. The peak corresponds to the creation of a crack at the location of maximum stress, which ultimately results in the breaking of the sample.

At the raw state, the NAO reaches a US of 29.4 MPa at 1.3% strain and the LS reaches 28.2 MPa at 2.2% strain. After exposition to heat, the NAO reaches 16.4 MPa at 0.9% strain and the LS reaches 14.8 MPa at 1.3% strain.

This plot shows that, although both raw materials have a very similar US, the elastic modulus of the raw LS is about 35% lower than that of the NAO. One can also see that the thermal exposition reduces the US about 45% for each material, and also reduces their elastic modulus about 20%. Finally, one can see that a major difference between the two formulations is their resilience. The LS is quite resilient, with a vast region of plastic behavior, both in the raw state and after exposition to heat, but the raw NAO is much more brittle, and the brittleness is even worse after exposition to heat.

In order to gain a better understanding of the mechanical behavior of the two formulations, we observe the crack faces of beams that were exposed to heat and flexure tested. The LS beam, shown in Figure 16a, features many millimeter-long fibers that protrude from the sample. These fibers correspond to the geometric description of the steel fibers. On the contrary, the crack face of the NAO beam, shown in Figure 16b, exhibits no visible fibers.

The results of the mechanical tests can be summarized as follows:
In the raw state:
○Both materials exhibit similar strength;○The NAO material is stiffer than the LS material;○The NAO material is more brittle than the LS material.
After exposition to high temperature:
○The materials both lose about 45% of their strength;○The materials both lose about 20% of their stiffness;○The LS material keeps its resilience, but the NAO material becomes even more brittle.


## 4. Discussion

During the severe braking tests, the LS-GCI configuration shows better behavior regarding wear and integrity than the NAO-GCI configuration. The NAO pad wears very quickly and cannot reach the end of the test. It suffers a dozen deep cracks.

The SEM observations and EDS analyses show a layer of solid oxidized iron at the surface of the LS pad. On the surface of the NAO pad, the particles appear coarse, and the usual structures of primary-secondary plateaus with finely ground and compacted particles can hardly be seen. The behavior of these two configurations can be described following Berthier’s definition of wear [3], which states that two steps are needed for wear to happen in a brake system:Material must detach from the first bodies (source flow).Detached material must be ejected from the system permanently (wear flow).

Here, the two configurations (LS-GCI and NAO-GCI) undergo different mechanisms of detachment and ejection. The measured values of wear rate suggest the ejection rate is far lower for the LS-GCI configuration than it is for the NAO-GCI configuration.

Regarding the detachment from the first bodies, material characterization coupled with test data can be used to interpretate the involved mechanisms. It was shown that exposition to high temperature damages both materials by similarly reducing their ultimate strength. The raw NAO material is also less resilient than the raw LS material, and this difference is increased by thermal damage. In the meantime, the diffusivity of the LS pad material in the normal direction is about 2.6 times higher than that of the NAO pad material, and the LS-GCI configuration produces a friction 30–40% higher than the NAO-GCI configuration.

At the LS-GCI interface, the high friction generates a significant amount of heat, but the relative high diffusivity helps mitigate the surface temperature, which reduces the material damage. Moreover, steel fibers are able to melt and create a film of molten metal. This film shears to accommodate speed differences and reduces stress localizations in the interface, thus, protecting the underlying material from large amounts of material detachment. At the NAO-GCI interface, less heat is generated due to lower friction, but the low diffusivity favors heat buildup and a large increase in surface temperature. This high temperature reduces the strength and the resilience of the material, thus, allowing for more detachment of debris. It appears that wear debris cannot be trapped in the contact, probably due to the very high temperature which rapidly damages the matrix and does not allow the mineral fibers to play their role as particle traps (they should act as anchors planted in the matrix, obstructing the flow of debris, which is not allowed here given the level of stress).

The high sliding speed may also play a major role in the ejection of debris. In a typical braking interface, detached debris are milled by the contact solicitations. The longer they are in contact, the finer the particles become. Their fine size allows them to compact into plateaus that are sheared to accommodate speed differences. At the NAO-GCI interface, a large amount of detached debris is quickly driven towards the outlet of the contact by the high speed without any chance of being finely milled. The result is that little to no compacted plateaus are created, and the ejected flow (wear flow) is high. At the LS-GCI interface, the film of molten metal is sheared to accommodate speed.

Concerning the creation of cracks, they are most likely caused by thermo-mechanical solicitations. In particular, the large temperature gradient relative to depth (around 200 °C in 3.9 mm) could be responsible for large amounts of differential expansion between the surface and the bulk, thus, creating large stresses along the longer direction of the pad. The values of such stresses were not calculated, as this would require setting up a finite element model, fed by realistic models for the two pad formulations. This calculation could be done in a future study.

The mechanical tests show that both formulations lose nearly half their mechanical strength when exposed to high temperature in air atmosphere. This loss is certainly due mainly to the thermal damage of the organic resin matrix and the aramid fibers. A major difference is the plastic behavior of the two formulations. The LS material shows a large area of plastic deformation, but the NAO material is rather brittle, and even more so after thermal damage. The worn LS pads do not show macroscopic cracks despite being exposed to strong heat longer than the NAO pads during braking. This suggests that the resilience of the materials plays a significant role in the creation of large cracks. On the one hand, small cracks may appear in the LS pad, but its high resilience prevents them from growing too much. On the other hand, when cracks appear in the NAO pad, in which mechanical behavior is brittle and even more so after thermal damage, they can grow easily, so much so that it represents an issue regarding the integrity of the whole pad.

The brittleness of the material could be attributed to the resin, as phenolic resins are known to be brittle. In the LS formulation, the long steel fibers mitigate this behavior and give more toughness to the composite. However, in the NAO formulation, the short mineral fibers struggle to achieve this function. This scenario is supported by existing results showing that braking materials with shorter mineral fibers show poor toughness [32].

## 5. Conclusions and Future Work

The severe braking tests conducted in this study were able to discriminate a low-steel and a NAO-pad formulation sliding against cast iron discs. The two configurations generated different CoF values, (which were 30–40% higher with the LS-GCI), but more importantly, they showed major differences regarding the wear and integrity of the brake pads. Notably, while the LS pads show very high amounts of wear (up to 9–30 µm/s) as compared with typical braking values, the NAO pads wear much faster (up to 40–80 µm/s) and exhibit a dozen major cracks. Surface observations of the worn parts, as well as toughness tests conducted on the pad materials, allowed us to better understanding the mechanisms involved, with respect to the content of the two formulations. The steel fibers seem to play several crucial roles in the preservation of the LS material; when encountering severe braking conditions, they melt and create a film of steel that can protect the base material. Secondly, they increase the resilience of the material, and impede the growth of large cracks when the material undergoes internal thermo-mechanical stresses caused by the braking conditions. These long fibers are able to maintain this resilience when the base material is thermally damaged. Conversely, at this level of solicitation (high temperature), the short mineral fibers do not seem to be able to achieve these functions and allow for critical wear and loss of integrity.

Future studies could follow several directions. One approach would be to conduct a study to further understand the mechanisms leading to dramatic wear in NAO pads without metal constituents submitted to severe braking conditions. In particular, investigating the seemingly poor capacity of plateaus of compacted third body to settle and accommodate tribological solicitations would be an interesting study. Further investigation of the thermo-mechanical stresses could be done by finite element analysis, based on measured properties of the pad materials, in order to better understanding the parameters initiating cracks and driving their growth.

Another approach for future advances would be to study designing non-metal NAO brake pads that could meet the challenges of these severe braking conditions. Such studies could propose to change the shape and size of mineral components, with the purpose of improving their tribological and mechanical contribution. Finally, a different approach would be to improve other aspects of the formulations, such as the phenolic resin matrix. A more resilient, more thermal resistant matrix could improve the performance of the brake pad materials.

## Figures and Tables

**Figure 1 materials-14-05531-f001:**
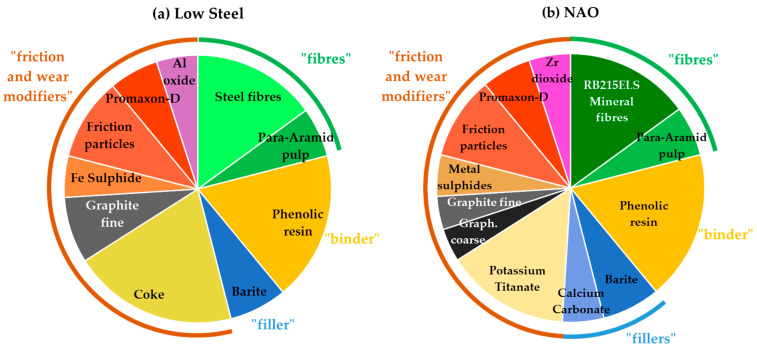
Composition of the two formulations in volume fraction: (**a**) LS formulation; (**b**) NAO formulation.

**Figure 2 materials-14-05531-f002:**
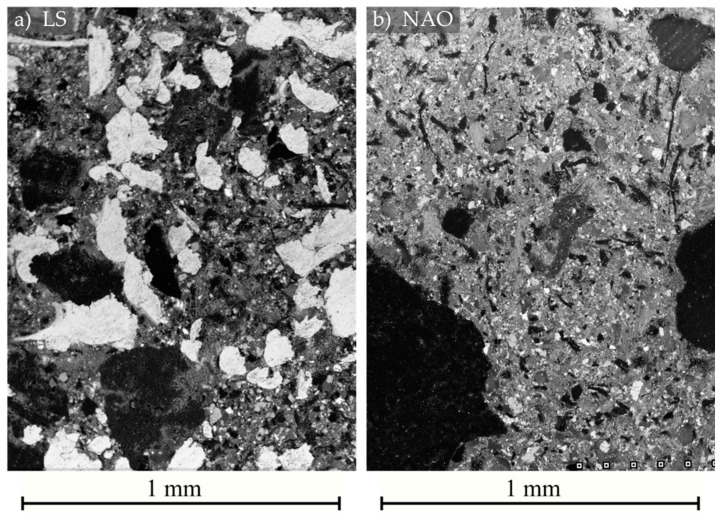
Microstructure of the materials: (**a**) LS formulation; (**b**) NAO formulation. SEM backscattered electron micrography.

**Figure 3 materials-14-05531-f003:**
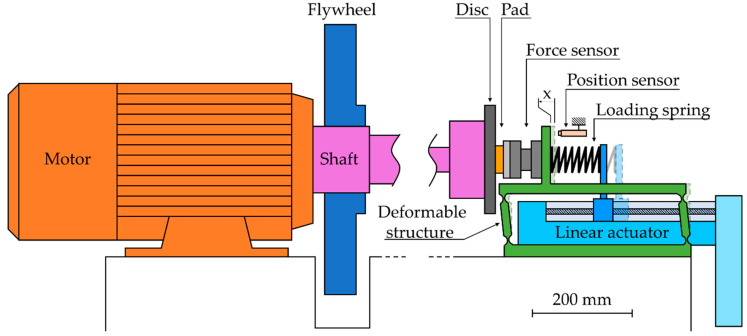
Simplified schematic of the test setup in sideview.

**Figure 4 materials-14-05531-f004:**
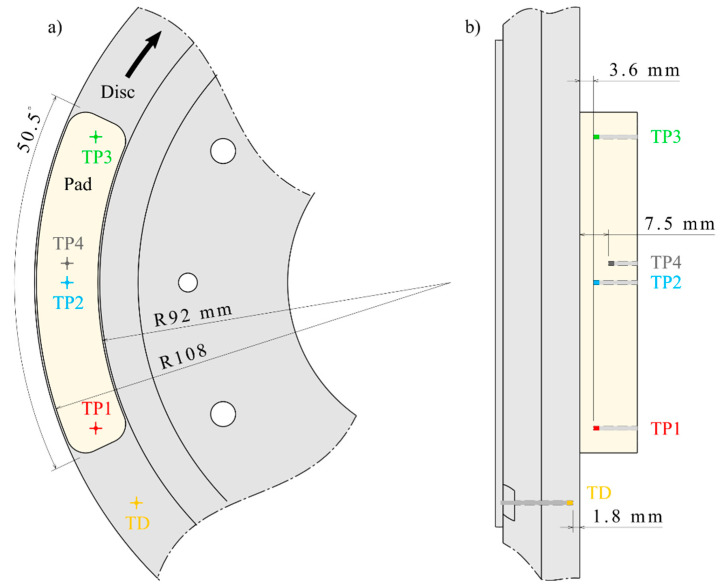
Details of the friction parts’ geometry and temperature measurements: (**a**) Front view; (**b**) side view.

**Figure 5 materials-14-05531-f005:**
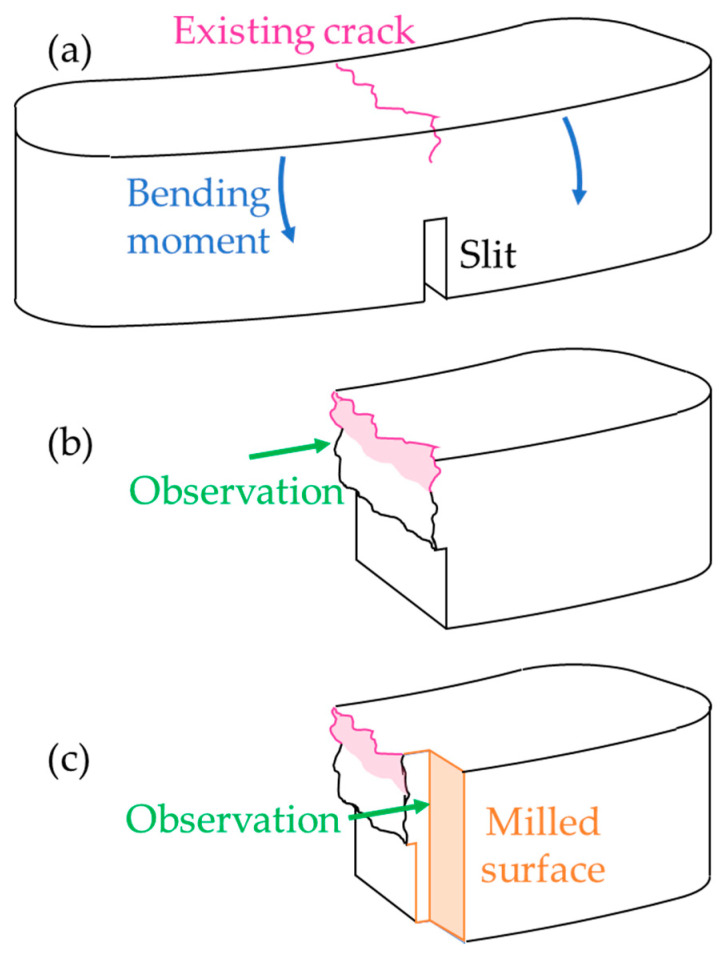
Procedure for investigating crack characteristics: (**a**) Position of the breaking slit; (**b**) observation of the crack face; (**c**) milling enabling the observation of the material near the crack.

**Figure 6 materials-14-05531-f006:**
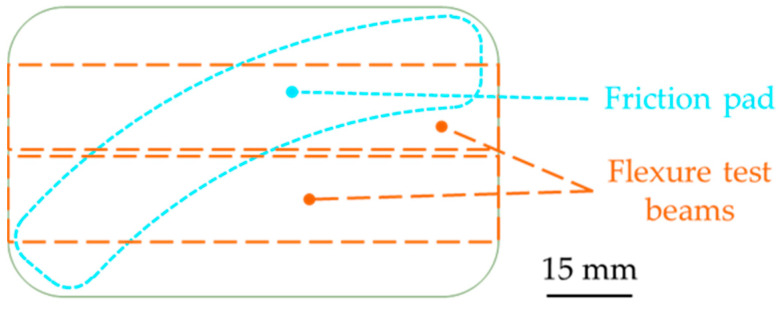
Sampling geometry for one friction sample or two toughness test samples out of an original molded lump.

**Figure 7 materials-14-05531-f007:**
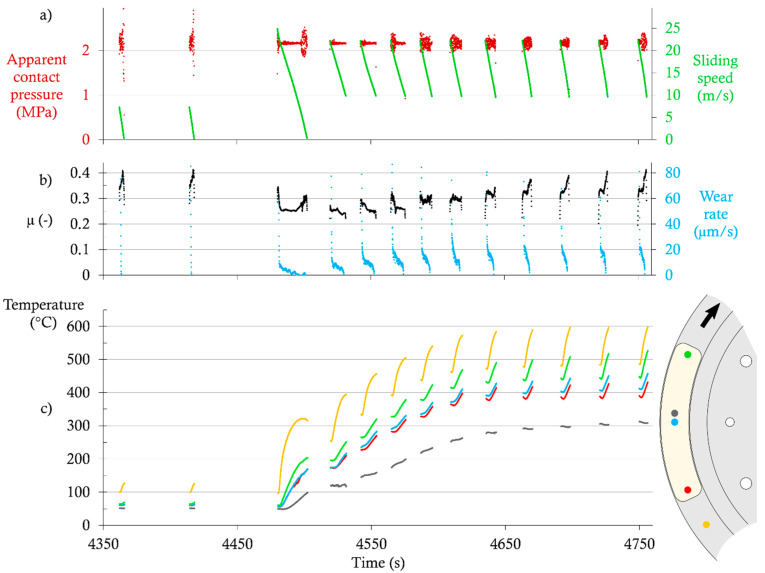
Extract of test data for the LS-GCI configuration: (**a**) Apparent contact pressure and sliding speed; (**b**) coefficient of friction (CoF) and wear rate; (**c**) disc and pad temperatures.

**Figure 8 materials-14-05531-f008:**
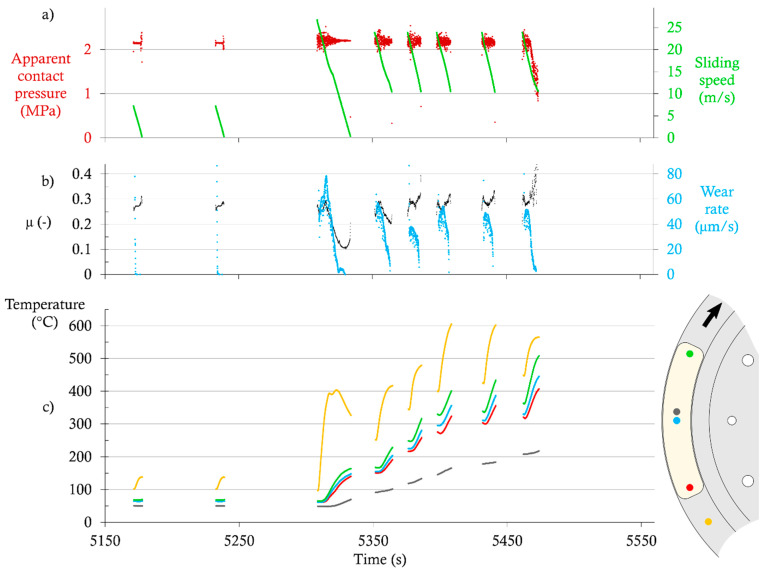
Extract of test data for the NAO-GCI configuration: (**a**) Apparent contact pressure and sliding speed; (**b**) coefficient of friction (CoF) and wear rate; (**c**) disc and pad temperatures.

**Figure 9 materials-14-05531-f009:**
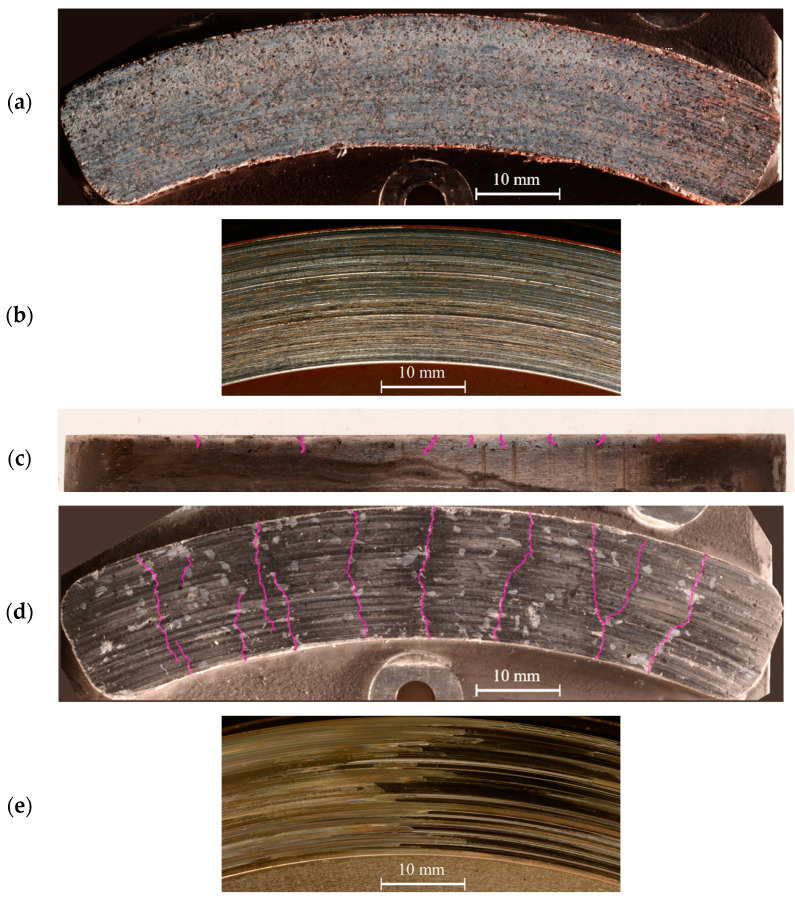
Worn surfaces: (**a**) Surface of an LS pad; (**b**) corresponding GCI disc; (**c**) sideview of a NAO pad; (**d**) surface of the NAO pad; (**e**) corresponding GCI disc. The purple lines highlight the paths of cracks visible to the naked eye.

**Figure 10 materials-14-05531-f010:**
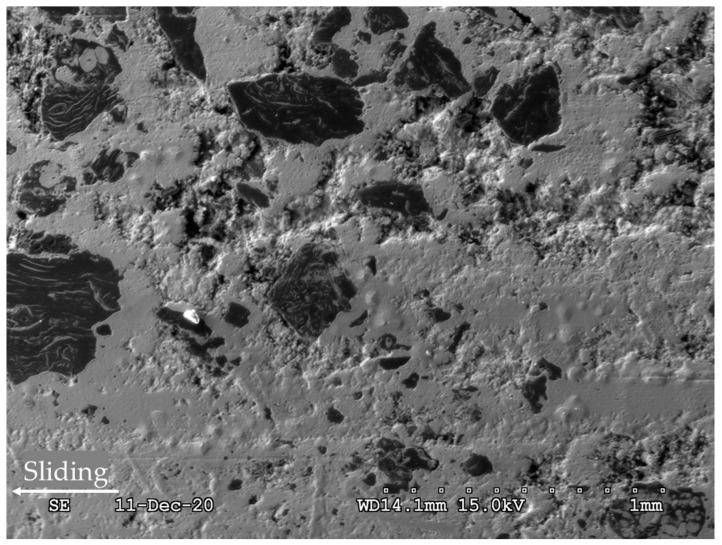
Worn surface of an LS pad. SEM secondary electrons image.

**Figure 11 materials-14-05531-f011:**
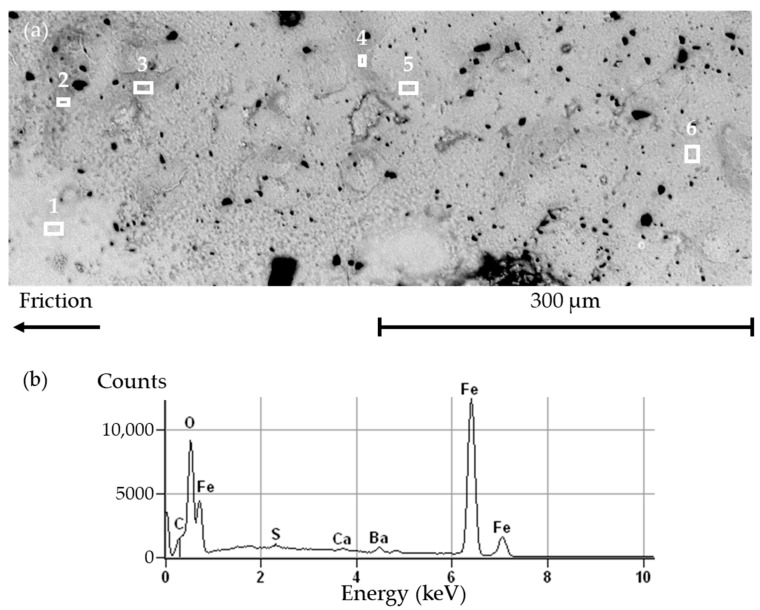
Worn surface of an LS pad: (**a**) SEM backscattered electron image; (**b**) EDS spectrum representative of analyses 1 to 6 (accelerating voltage 20 kV, working distance 14.1 mm).

**Figure 12 materials-14-05531-f012:**
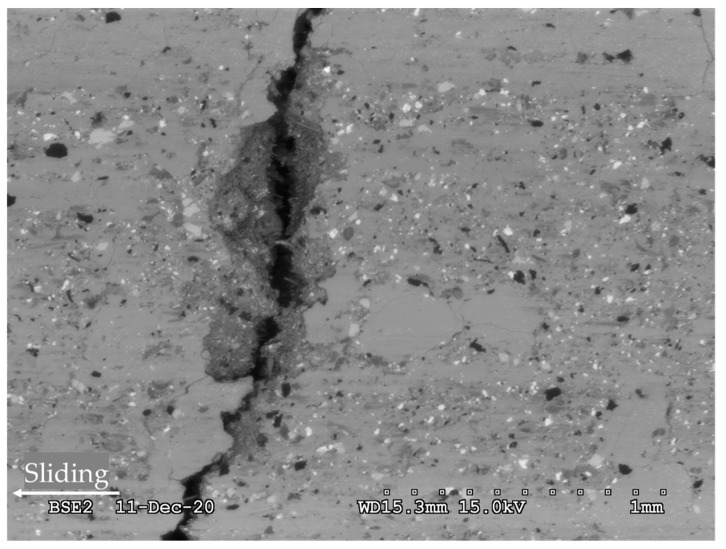
Worn surface of an NAO pad. SEM backscattered electron image.

**Figure 13 materials-14-05531-f013:**
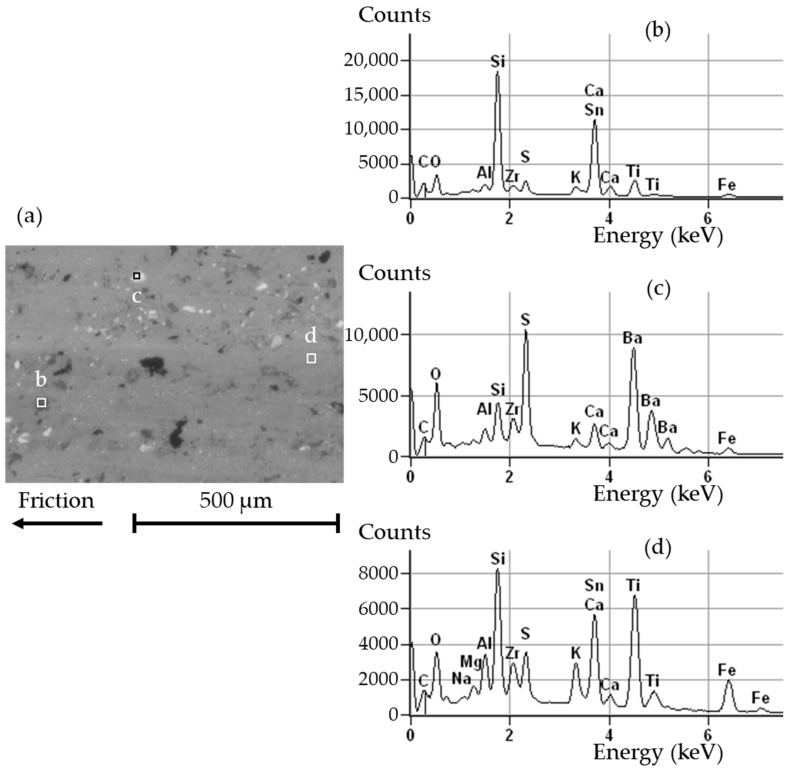
Worn surface of an NAO pad: (**a**) SEM backscattered electron image; (**b**–**d**) EDS spectra corresponding, respectively, to the areas labeled as b, c, and d in (**a**) (accelerating voltage 20 kV, working distance 15.3 mm).

**Figure 14 materials-14-05531-f014:**
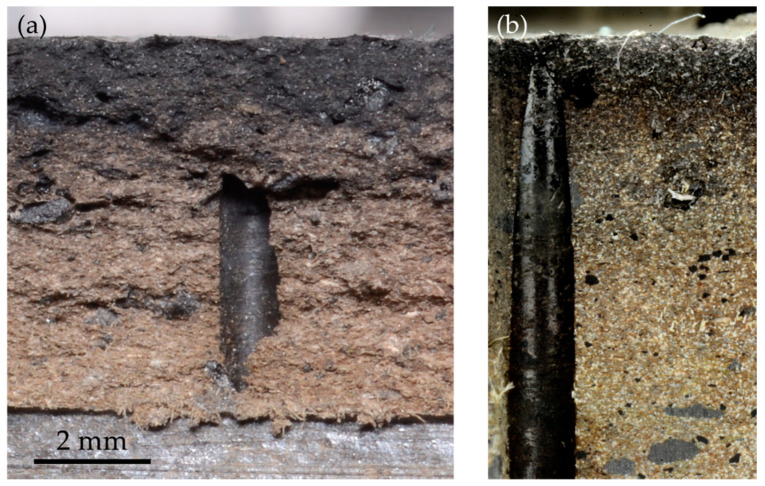
Observation of the face of a crack in a NAO pad: (**a**) Face of the crack; (**b**) aspect of the material at the vicinity of the crack.

**Figure 15 materials-14-05531-f015:**
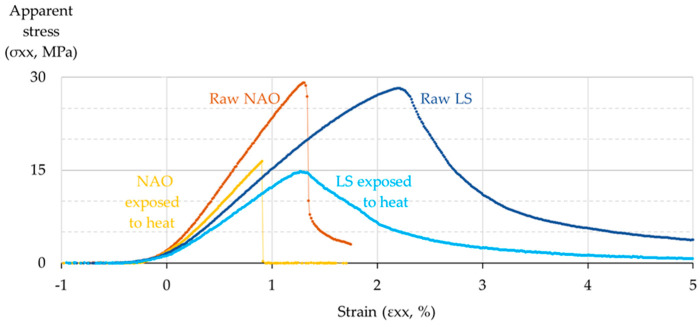
Graph of stress and strain in three-point flexure for the NAO and the LS materials, in the raw state and after exposition to heat.

**Figure 16 materials-14-05531-f016:**
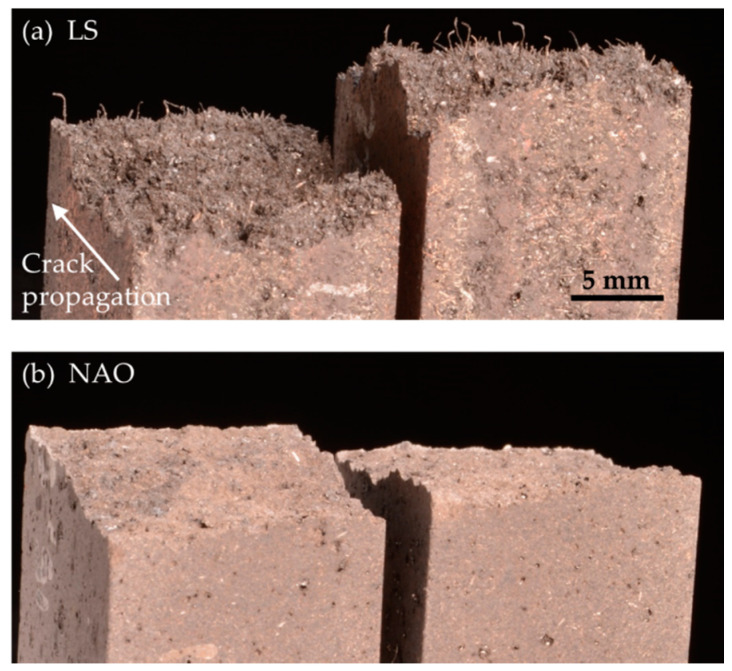
Flexure test samples after rupture: (**a**) LS beam after exposition to heat and flexure test; (**b**) NAO beam after exposition to heat and flexure test.

**Table 1 materials-14-05531-t001:** Geometry of steel fibers and RB215ELS mineral fibers.

Constituent	Average Diameter (µm)	Average Length (µm)
Steel fibers	50	1800
RB215ELS mineral fibers	5.5	150

**Table 2 materials-14-05531-t002:** Diffusivity values of the pad materials along the direction normal to their surfaces.

Material	Thermal Diffusivity at 20 °C Direction Normal to Friction Surface (mm^2^·s^−1^)
LS	0.988
NAO	0.376

**Table 3 materials-14-05531-t003:** Braking test program at a reduced scale.

Sequence	Number of Applications	V_sliding_ (±1 m·s^−1^)	P_contact_ Apparent (MPa)	Simulated Inertia (±0.1 kg·m^2^)	T_initial_ Disc (°C)
Run-in	100	from 7.3 to 0.4	2.0	4.1	100
High-speed braking	1	from 26 to 0.5	2.0	3	100
High-temperature braking	20	from 23 to 10	2.0	3.5	Imposed, from 250 to 500

## Data Availability

The data presented in this study are available on request from the corresponding author.

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
