# Peer review of "Differences in Wear and Material Integrity of NAO and Low-Steel Brake Pads under Severe Conditions"

_materials, 2021, doi:10.3390/ma14195531_

Round 1

Reviewer 1 Report

Wear performance is important to the brake pads materials. Gray cast iron is widely used as brake pads materials. Authors have studied friction properties for two prototype formulations of NAO and LS. Some results have been present. However, the current form of this study cannot be acceptable. Some aspects as listed below: 

1. In section 2.1 Friction Materials, more details of the grey cast iron should be given, such as mechanical properties?  Besides, in Figure 1, why the two formulations in volume fraction of LS and NAO are used? More details should be given. 

2. Which factor has the greatest impact on friction performance in the two formulations?

3. What is the standard for the braking tests?

4. In Table 2, how to control the Vsliding?

5. In Fig.6 and Fig.7, how to measure the wear rate and contact pressure? Why the time is different in Fig.6 and Fig.7?

6.  Here, is the friction coefficient an average value?

7. The discussion is not sufficient in the paper. More mechanistic analysis can be given in the paper.

8. It is recommended that the wear and friction performance for a feasible material formulation of brake pad should be optimized.

Author Response

Q1. In section 2.1 Friction Materials, more details of the grey cast iron should be given, such as mechanical properties?  Besides, in Figure 1, why the two formulations in volume fraction of LS and NAO are used? More details should be given. 

Response: Regarding the cast iron, the following correction is added to the article at lines 107-108:

“The discs are made of grey cast iron (GCI) EN-GJL-250, with yield strength of 250 MPa (as per the standard) a measured hardness of 180 HV”

Regarding the pad formulation, we propose to show them in volume fraction as this unit highlights several similarities between the two formulations. It also makes it more comfortable to compare the formulation with surface observations which are also bringing volume information.

the following sentence is added to the article at lines 115-116:

“Volume fractions are chosen as they make comparison with surface observations easier than mass fractions.”

Q2. Which factor has the greatest impact on friction performance in the two formulations?

Response: The effect of different parameters on the friction force was not considered in the article, as the main topic is the severe wear and the loss of integrity of the pads.

Q3. What is the standard for the braking tests?

Response: Several standard tests (W500, AMS, AK-M and HSF) were done during the full-scale preparatory work for this study. It is found that these general tests have too many parameters variations, which impede the fine scientific understanding that we were looking for. This has been clarified by adding the following sentence to the articles at lines 96 to 99:

“Unlike general-purpose standard braking tests, the design of a tailored test program, with a focus on scientific understanding, allows us to keep only a limited set of desired parameters and a well-controlled solicitation range.”

Q4. In Table 2, how to control the Vsliding?

Response: Questions 4, 5 and 6 are answered together by adding a paragraph to the article at lines 161 to 186

Q5. In Fig.6 and Fig.7, how to measure the wear rate and contact pressure? Why the time is different in Fig.6 and Fig.7?

Response: Questions 4, 5 and 6 are answered together by adding a paragraph to the article at lines 161 to 186

Q6.  Here, is the friction coefficient an average value?

Response: Questions 4, 5 and 6 are answered together by adding a paragraph to the article at lines 161 to 186

Q7. The discussion is not sufficient in the paper. More mechanistic analysis can be given in the paper.

Response: We agree that the discussion was too short. It has been enriched by adding the precisions to the mechanism causing differences in the amount of wear. This text is supported by precisions and references added in the introduction regarding tribological mechanisms.

The text added to the introduction in lines 35 to 48 is reported below:

“The braking contact is composed of two first bodies (disc and pad) and an intermediate layer, called the third body [1]. The third body performs three main functions: bearing the contact loads, accommodating the speeds, and ensuring the separation of the first bodies. The third body particles travel on the surface of the first bodies and form the tribological circuit [2], [3]. Various flows of particles are likely to be activated in this circuit, in particular:

  • Material detached from the first bodies constitutes the main source flow;
  • Particles that circulate inside the contact while accommodating velocities constitute the internal flow;
  • Particles that leave the contact permanently constitute the wear flow.

Primary and secondary contact plateaus form at the interface [4]. The primary plateaus are constituted by the wear resistant constituents of the pad (such as steel fibers or ceramic particles) and form nucleation sites for the secondary plateaus, which are created by detached particles, compacting in front of the primary plateaus.”

The text added to the Discussion in lines 434 to 445 is reported below:

“Our interpretation is that the steel of the pad fibers melts due to the braking energy and creates a very thin film of liquid metal, that acts as a third body layer, by bearing the loads and circulating in the contact to accommodate the sliding speed. The wear occurs when this material leaves the contact, and thus the system, due to very high speed. The NAO pad does not contain steel, so this mechanism cannot take place. Moreover, the more usual structures of primary-secondary plateaus with finely ground and compacted particles can hardly be seen on the surface of the NAO pad. Usually, the longer the particles stay inside the interface, the finer they are ground by contact loads. Our interpretation is that because of the very high sliding speed, the particles detached from the first bodies are moved quickly towards the exit of the contact. Here, the particles appear coarse, confirming their short stay. Their large size and their speed impede them from compacting into usual third body plateaus. The result is that these large particles, moved out of the contact at a quick pace represent large wear flow.”

A short precision concerning the stresses in the pads has been added in lines 450 to 452.

Q8. It is recommended that the wear and friction performance for a feasible material formulation of brake pad should be optimized.

Response: The materials presented in the study are very close to industrial materials, so the authors believe such materials are feasible for production and achieve realistic performance for brake pads.

Reviewer 2 Report

The article was written in a clear and correct manner. The paper is consistent and deserves recognition for the quality of preparation. The paper is divided into several main sections in which the research problem is thoroughly described. The paper can be published after some minor comments:

  1. Strongly suggest that the novelty of this paper be demonstrated in relation to other thematically similar research papers, preferably in the introduction.
  2. The introduction should be significantly expanded - only 13 works that are cited in the introduction appear in the paper. I suggest expanding the introduction to include works in the field of the presented problems, such as: 10.12913/22998624/62628, 10.1016/j.wear.2020.203539 and 10.1016/j.simpat.2018.01.009.
  3. Please provide a source for Figure 1.
  4. Please provide better quality for figure 6 and 7. 
  5. Please refer to the quantitative evaluation of the research findings in the discussion or conclusion section. 

Author Response

Please find attached a version of the article where every modification in the text appear in orange color.

Q1. Strongly suggest that the novelty of this paper be demonstrated in relation to other thematically similar research papers, preferably in the introduction.

Response: The authors fully agree with this suggestion. After adding new references to the introduction (see response to Q2), the novelty of the work is detailed in lines 95 to 103, by adding the following text:

“The use of a reduced scale tribometer allows for a very close monitoring and detailed understanding of the phenomena involved at the interface. Unlike general-purpose standard braking tests, the design of a tailored test program, with a focus on scientific understanding, allows us to keep only a limited set of desired parameters and a well-controlled solicitation range. Coupling such tests with multi-scale observations and analyses of worn surfaces, with observation of crack faces, as well as mechanical characterization of the materials constitutes a novel approach. This approach covers a continuum between brake system performance, tribological interface behavior, mechanical behavior of the parts and the design of the material formulations.”

Q2. The introduction should be significantly expanded - only 13 works that are cited in the introduction appear in the paper. I suggest expanding the introduction to include works in the field of the presented problems, such as: 10.12913/22998624/62628, 10.1016/j.wear.2020.203539 and 10.1016/j.simpat.2018.01.009.

Response: The introduction has been extended in several ways.

Adding the following state of the art on tribological mechanisms at the braking interface, in lines 35 to 48:

 “The braking contact is composed of two first bodies (disc and pad) and an intermediate layer, called the third body [1]. The third body performs three main functions: bearing the contact loads, accommodating the speeds, and ensuring the separation of the first bodies. The third body particles travel on the surface of the first bodies and form the tribological circuit [2], [3]. Various flows of particles are likely to be activated in this circuit, in particular:

  • Material detached from the first bodies constitutes the main source flow;
  • Particles that circulate inside the contact while accommodating velocities constitute the internal flow;
  • Particles that leave the contact permanently constitute the wear flow.

Primary and secondary contact plateaus form at the interface [4]. The primary plateaus are constituted by the wear resistant constituents of the pad (such as steel fibers or ceramic particles) and form nucleation sites for the secondary plateaus, which are created by detached particles, compacting in front of the primary plateaus.”

Adding the following state of the art on severe wear and loss of integrity of brake parts, in lines 56 to 79:

“The wear and integrity of brake components under severe conditions is studied in a variety of works. While some authors focus on very specific types of environment conditions [18], many authors study the cases of high energy and emergency braking. Most of these works focus on the loss of integrity by the apparition of cracks in gray cast iron discs (such as [19, 20]) and steel discs [21, 22]. These papers point-out the role of alternating tensile-compressive stress cycles favored by high temperature gradients and fast temperature changes in the disc.

Fewer works study excessive wear and loss of integrity of brake pads or shoes for drum brakes. Some papers focus on metal-matrix brake pads (such as [23]), designed for high speed trains or heavy-duty vehicles. Regarding organic matrix brake materials, Barros et al [24] point-out the role of contact pressure in the transition to severe wear. Laguna-Camacho et al [25] study drum shoes used in real service for eight months in warm environment and find major cracks that they attribute to fatigue from repeated impacts of the shoe against the drum. Elzayady and Elsoeudy [26] study three formulations of NAO pads in real service and in severe laboratory braking tests, and find cracks attributed to thermal fatigue. They note that the addition of graphite and barite lowers the generated heat and produces less cracks. Park et al [27] study the wear and airborne emission with several low-steel and non-steel pad formulations and find that the low steel generated more wear and more airborne particles. They also note that more airborne particles are emitted at higher velocity. Their fine surface observations enable them to draw schematic of the pads surfaces topography. Cong et al [15] focus on the role of the phenolic resin in their tests at high speed and moderate load. A pad material using their new resin modified with latex generates about 25% less wear than a pad with classic resin. It also keeps its integrity while the classic material suffers millimetric cracks.”

The authors wish to thank the reviewer for his kind proposition of three works that could be added to the references. doi:10.1016/j.wear.2020.203539 is of great value in positioning the work discussed here. It was added in the state of the art in lines 72 to 76. For 10.12913/22998624/62628 and 10.1016/j.simpat.2018.01.009, despite the quality of the works, the authors would prefer not to add these references. A careful reading and discussion between the authors led us to think that adding these two references would need to enlarge the introduction by adding many other works related to lift systems and also to numerical simulations of brake pad wear. For the sake of brevity, the authors would prefer not to add these references.

Q3. Please provide a source for Figure 1.

Figure 1 and the formulations it describes are the work of the authors. It has been explicated by modifying lines 87 and 88 with the following mention:

“We conduct a reduced scale experimental laboratory study with two prototype formulations, designed by the authors, aimed at representing usual NAO and LS materials.”

Q4. Please provide better quality for figure 6 and 7.

Response: The DPI resolution of the figures export has been doubled to increase quality.

Q5. Please refer to the quantitative evaluation of the research findings in the discussion or conclusion section.

Response: Values have been added in the conclusion section:

Lines 469-470: “The two configurations generated different values of CoF, (which was 30-40% higher with the LS-GCI)”

Lines 472-474: “Notably, while the LS pad shows very high amounts of wear (up to 9-30 µm/s) compared to usual braking values, the NAO wears much faster (up to 40-80 µm/s) and exhibits a dozen of major cracks”

Reviewer 3 Report

No comment

Author Response

Please find attached a version of the article where every modification in the text appear in orange color.

Round 2

Reviewer 1 Report

Wear performance is important to the brake pads materials. Gray cast iron is widely used as brake pads materials. Authors have revised the manuscript according to the commends. However, the current form of this study cannot be acceptable. Some aspects as listed below: 

1. In Figure 2, what is the resolution of the SEM image? The scale label under the picture needs to be improved for the specifications.

2. Which factor has the greatest impact on wear performance in the two formulations?

3. In Fig.9, what is the scale in (b) and (d)? 

4. The discussion is still not sufficient in the paper. EDS results should be given.

Author Response

The authors would like to thank Reviewer 1 for their thorough work on the manuscript reviewing and their very useful suggestions. Below are the point-by-point answers to the comment. A corrected version of the article is joined with the modified text in orange color for better visualization.

Wear performance is important to the brake pads materials. Gray cast iron is widely used as brake pads materials. Authors have revised the manuscript according to the commends. However, the current form of this study cannot be acceptable. Some aspects as listed below: 

  1. In Figure 2, what is the resolution of the SEM image? The scale label under the picture needs to be improved for the specifications.

Both images are presented at the same scale. In order to improve clarity and to remove every question, one scale bar has been placed under each picture of figure 2.

  1. Which factor has the greatest impact on wear performance in the two formulations?

Here the variation between the two formulations is focused on the fiber type. The characteristics of the fibers have the greatest impact on the wear and preservation against cracks. The length and the resilience of the steel fibers (“Low Steel” formulation) help preventing cracks, while their ability to melt when encountering high braking temperatures provides a lubricant film of molten material and prevent excessive wear. Mineral fibers (“NAO” formulation) don’t melt, because of their high temperature performance, but act as obstacle to trap the wear particle and form a third body layer by particle compaction. These points are stated in the discussion in lines 495-499 and lines 528-531, and are reminded and summarized in the conclusion at lines 541-549

  1. In Fig.9, what is the scale in (b) and (d)? 

All subfigures of figure 9 are presented at the same scale. In order to improve clarity and to remove every question, one scale bar has been in under each picture of figure 9 (b) and (d).

  1. The discussion is still not sufficient in the paper. EDS results should be given.

Results of EDS analyses have been added for the worn surfaces of LS and NAO pad. The results are given in figure 11 and 13 and described in lines 343-352 and lines 365-389.

The discussion has been extended with a detailed interpretation of the wear mechanisms (lines 472-507). In order to carry a finer analysis, the authors chose to use values of thermal diffusivity, which were added in Table 2, introduced in lines 144-145.
